# Prognosis of colorectal cancer in Tikur Anbessa Specialized Hospital, the only oncology center in Ethiopia

**Eyob Kebede Etissa**[1]\*, **Mathewos Assefa**[2], **Birhanu Teshome Ayele**[3]

**1** GAMBY College of Medical and Business Sciences, Addis Ababa, Ethiopia, **2** Oncology Department, School of Medicine, Addis Ababa University, Addis Ababa, Ethiopia, **3** Faculty of Medicine and Health Sciences, Division of Epidemiology and Biostatistics, Stellenbosch University, Stellenbosch, South Africa

\* eyobke@gmail.com

## Abstract

### Introduction

Colorectal cancer is the third most commonly diagnosed cancer in males and the second in females worldwide. According to the Addis Ababa cancer registry, it is the first in male and fourth in female in Ethiopia. However, there have not been studies on prognostic factors and survival of colorectal cancer. Hence, this study aimed to estimate survival time and identify prognostic factors.

### Methods

In this institution based retrospective study, medical records review of 422 colorectal cancer patients and telephone interview was used as sources of data. Survival time was estimated using Kaplan-Meier estimator. Prognostic factors were identified using the multivariable Cox regression model.

### Results

Patients diagnosed with rectal cancer had 76% (HR: 1.761, 95% CI: 1.173–2.644) increased risk of dying compared to colon cancer patients. Node positive patients were 3.146 (95% CI: 1.626–6.078) times likely to die compared to node-negative and metastatic cancer were 4.221 (95% CI: 2.788–6.392) times likely to die compared to non-metastatic patients. Receiving adjuvant therapy reduced the risk of death by 36.1% (HR: 0.639 (95% CI: 0.418–0.977)) compared to patients who had an only surgical resection. The median survival time was 39 months and the overall five years survival rate was 33%.

### Conclusions

The overall survival rate was low and a majority of the patients were young at presentation. Patient's survival is largely influenced by the advanced cancer stage at presentation and delays in the administration of adjuvant therapy. Receiving adjuvant therapy was among the good prognostic factors.

**Data Availability Statement:** All relevant data are within the manuscript and its Supporting Information files.

**Funding:** The author(s) received no specific funding for this work.

**Competing interests:** The authors have declared that no competing interests exist.

## Introduction

Cancer is the leading cause of death globally, surpassing mortality rates of tuberculosis, malaria and HIV/AIDS combined, and it is quietly taking centre stage. Colorectal cancer is a cancer of the large intestine. Anatomically, it also is known as colon cancer or rectal cancer but when both present with similar features they are termed as colorectal cancer (CRC) [1–5]. Clinical presentation of CRC depends on its size, presence or absence of metastases and tumour location. Early CRC often has no symptoms [6]. CRC is the third most commonly diagnosed cancer in males and second in females, making the disease the second-most common cause of cancer-related death worldwide and remains as one of the biggest killer cancers in the world. Its burden is expected to increase. Despite this increasing burden, in Africa CRC continues to receive a relatively low public health priority due to limited resources and more other prompt diseases [3,5,7,8].

In Ethiopia, CRC is the third most prevalent cancers among the entire adult population and patients often present with advanced stages of cancer [9]. According to the Addis Ababa cancer registry, it is the first in males and fourth in females [10]. From 14,500 cancer-related deaths among males, 11.2% were due to CRC and 4.8% of 26,200 cancer-related deaths in women were due to CRC [11]. Currently, CRC is the most common cancer representing 13% of all malignant tumours in the gastrointestinal tract. More CRC deaths (52%) occur in the less developed regions of the world, reflecting poorer survival in these regions [3,12].

Various studies have shown that CRC survival is mainly determined by age at diagnosis, complicated colorectal cancer (perforation, obstruction, bleeding), tumour site, tumour grade, tumour maturity, tumour size, lymph node involvement, distant metastasis, treatment modalities and complication after treatment. Some literature reported that sex, educational status, family history, pre-operative carcinoembryonic Antigen (CEA) level and lymphovascular invasion as significant prognostic factors [13–16]. The highest five-year survival rates (72%) were in Israel and North Korea, in North America, Europe and Australia/New Zealand survival varies from 65%-70%, 55% in other developed countries, 39% in developing countries including India and 14% in Africa [6,17,18].

In Africa, data for CRC survival is scanty. In Ethiopia, to the best of our knowledge, there is no study on prognostic factor and survival of colorectal cancer. For these reasons, this study explored the survival and prognostic factors of colorectal cancer patients in Tikur Anbessa Specialized Hospital (TASH) of Addis Ababa, Ethiopia.

## Materials and methods

A retrospective study was conducted on colorectal cancer patients referred to TASH Radiotherapy centre from January 2012 to December 2016. TASH is the only oncology and cancer referral centre providing services for the majority of cancer cases from all over the country [19].

The medical records of 422 colorectal cancer patients were reviewed and telephone follow-up was used to find out the vital status of patients. The follow-up time was from the first date of confirmed diagnosis to the date of death, date of loss to follow up or date of the last contact. Patients with unknown vital status or alive patients during last contact (phone) were considered as censored. Thirteen patients not meeting the inclusion criteria were excluded from the study. Data on socio-demographic characteristics, genetic factor (family history of CRC), comorbidities (HIV/AIDS, diabetic mellitus and hypertension), and pathological characteristics (primary tumour site, histopathology, tumour grade, size of the tumour, lymph node metastasis, distant metastasis, lymphovascular invasion, perineural

invasion and pre-operative CEA level and treatment modality) were extracted from the medical records.

## Sample size consideration

Sample size determination started by determining d (number of events) because of the number of observed events matters beyond the total number of patients in survival data analysis. The required total number of events (Freedman 1982) was calculated using [20]:

$d = \left( Z_{\alpha/2} + Z_{\beta} \right)^2 \frac{4}{In\Delta^2}$, assuming that $\Delta$ = hazard ratio is constant in time. For the two-sided log-rank test ($H_0$: $\Delta = 1$ $vs$ $H_A$: $\Delta \neq 1$).

Then, the required total number of patients can be calculated as $\frac{d}{Probablity\ (of\ an\ event)}$. A previous similar study conducted in Ghana showed that survival function (s (t)) = 0.16, thus we used $s$ ($t$) = 0.16. We also know that $\Delta = -\log(s(t)) = \log(0.16) = 1.832$.

To obtain the total sample size required to yield a power of 90%, a hazard ratio of 1.832 and 5% two-sided level of significance, the required number of events is:

$$d = \left( Z_{\alpha/2} + Z_{\beta} \right)^2 \frac{4}{In\Delta^2} = (1.96 + 1.282)^2 \frac{4}{In(1.832)^2} = 115.$$

$Z_{\alpha/2}$ = 1.96 for 95% confidence and $Z_{\beta}$ = probability of committing type II error to give power of 90%.

Accordingly, $n = \frac{d}{Probablity\ (of\ an\ event)} = \frac{115}{0.84} = 137$.

Adjusting for a loss of 20%. $n_{adj} = \frac{n}{1-loss} = \frac{137}{1-0.2} \approx 172$.

The required sample size was 172. However, all colorectal cancer patients treated in TASH from 2012 to 2016 and who fulfil the inclusion criteria were included in the study. Hence the sample used for this study was 422.

## Analysis plan

The analysis was performed using SPSS version 25 and Stata version 14. Descriptive analysis was done using frequency, percentages and median. Survival time was estimated using the Kaplan-Meier method. A log-rank test was used to compare (overall) survival of two or more groups and identify potential prognostic variables. In the univariate analysis, variables with p < 0.25 were entered into the multivariable Cox regression model. Deviation from the proportional hazard assumption of the Cox regression model was examined using plots of scaled Schoenfeld residuals, global test and log-log transformation of Kaplan- Maier survival curves of categorical variables. Prognostic factors of colorectal cancer survival were identified using multivariable Cox regression model. P-value of less than 0.05 was considered as statistically significant.

## Ethical approval and consent to participate

To access the medical records ethical approval letter for the proposed study was obtained from GAMBY Medical and Business Sciences College institutional research ethics review committee (GCA.A 20/2010) to Tikur Anbessa Specialized Hospital. Permission letter was obtained from Tikur Anbessa Specialized Hospital to extract data from medical records. Written consent could not be obtained as the patients were not at the facility. Verbal informed consent was obtained for patient's eligible for the phone interview, witnessed by two experienced data collectors who are not a part of the study and documented via written note. Minors were not included in the study.

## Results

### Summary of baseline information

A total of 422 colorectal cancer patients treated in Tikur Anbessa Specialized Hospital from January 2012 to December 2016 were considered in this retrospective study. One hundred seventy-five (41.5%) of the patients died in the study period while the remaining 247 (58.5%) were censored. Majority of the patients, 262 (62.1%) were male. The overall median age was 46 years with Interquartile range (IQR) of 23 and 185 (43.9%) were in the age group of 40–59 years. Fifty-two (12.3%) patients comorbidities 28 (6.6%) of the patients were hypertensive, 15 (3.6%) were diabetic and 7(1.7%) were HIV positive at presentation. The details are presented in Table 1.

As depicted in Table 2, colon, 217 (51.4%), and rectum, 144 (34.1%), were the common cancer sites involved. Forty-five (31.3%) rectal cancer patients had adjuvant chemotherapy or radiotherapy after surgery, 31 (21.5) had adjuvant chemotherapy and radiotherapy after surgery and 23 (16) did not receive any treatment. Majority of the patients, 333 (∼79%), have adenocarcinoma NOS, followed by mucinous adenocarcinoma (12.9%) and signet ring adenocarcinoma (4.7%). The majority of patients, 198 (46.9%), had well-differentiated adenocarcinoma.

Majority of the patients were diagnosed with a primary tumour of T3 (46.7%). About one-fourth patients presented with pathologically confirmed regional lymph node involvement. In 57.1% of cases, the lymph node status is unknown which means we do not know the TNM staging in this group of patients. Nearly half, (49.3%), of the patients, had received adjuvant chemotherapy or radiotherapy after surgery. All patients who underwent adjuvant therapy had surgery.

### The yearly incidence of death and survival of colorectal cancer patients

Among the total of 422 patients, 175 (41.5%) died and 144 (34.1%) were alive by October 2018, when patients status was last updated. The vital status of the remaining, 103 (24.4%), was unknown due to refusal to respond to calls and no working telephones. Overall, there was 822.25 person-years observation with a median follow-up time of 19 months (IQR 28). This makes the incidence rate of death of colorectal cancer 21.28% per 100 person-years observation.

The graph in Fig 1 shows the overall survival curve of colorectal cancer patients in TASH. The graph shows that patients' survival decreases as the observation time increases. One, three and five years overall survival rate were 80%, 55%, and 33% respectively. The median overall survival time was 39 months.

Table 3 shows the results of the Log-rank test of socio-demographic characteristics and the presence of comorbidities in CRC patients. No statistically significant differences were observed among the survival groups (P>0.05) although having comorbidities seemed to lower patients' survival rate.

The median survival time, survival rate and log-rank test for pathological and clinical variables were assessed univariately.

As shown in Table 4, pathological and clinical factors like primary tumour site, histopathology, tumour grade, size of the tumour, lymph node metastasis, distant metastasis and treatment modality were significantly associated with patients' survival time (p<0.05). Colon and colorectal cancer patients had better survival compared to that of the other commonest cancer site rectum. Patients with poorly differentiated tumour grade were at increased risk of death compared to those with well and moderately differentiated tumour grades. The median

**Table 1. Socio-demographic characteristics and co-morbidity history of colorectal cancer patients (n = 422) in Tikur Anbesa Specialized Hospital (TASH), Addis Ababa, Ethiopia, 2012–2016.**

| Variables | n (%) |
|---|---|
| **Age** (Median, IQR) 46 (23) | |
| ≤ 29 | 50 (11.8) |
| 30–39 | 79 (18.7) |
| 40–49 | 94 (22.3) |
| 50–59 | 91 (21.6) |
| 60–69 | 77 (18.2) |
| ≥70 | 31 (7.3) |
| **Sex** | |
| Male | 262 (62.1) |
| Female | 160 (37.9) |
| **Place of residence** | |
| Addis Ababa | 187 (44.3) |
| Out of Addis Ababa | 235 (55.7) |
| **Educational status** | |
| Unable to read and write | 16 (3.8) |
| Able to read and write | 2 (0.5) |
| Elementary (1–8) | 20 (4.7) |
| High school (9–12) | 19 (4.5) |
| College and above | 22 (5.2) |
| Unspecified | 343 (81.3) |
| **Family history of CRC** | |
| Yes | 1 (0.2) |
| No | 132 (31.3) |
| Unspecified | 289 (68.5) |
| **Comorbidities** | |
| Yes | 52 (12.3) |
| No | 370 (87.7) |
| **HIV/AIDS** | |
| Yes | 7 (1.7) |
| No | 415 (98.3) |
| **Diabetic mellitus** | |
| Yes | 15 (3.6) |
| No | 406 (96.2) |
| Unspecified | 1 (0.2) |
| **Hypertension** | |
| Yes | 28 (6.6) |
| No | 394 (93.4) |

IQR = inter-quartile range, HIV/AIDS = human immune virus/acquired immune deficiency syndrome.

survival time was higher (60 months) among those node-negative patients compared to those diagnosed with node-positive (24 months). Similarly, metastatic cancer patients had lower 5-year survival rate (2%). According to treatment modality, patients who received adjuvant chemotherapy and adjuvant radiotherapy after surgery and who received adjuvant chemotherapy or adjuvant radiotherapy after surgery had higher median survival time (73 months) and (46 months) respectively. Lower median survival time (6 months) was observed among those

**Table 2. Pathological and clinical characteristics of colorectal cancer patients (n = 422) in TASH, Addis Ababa, Ethiopia, 2012–2016.**

| Variables | | n (%) |
|---|---|---|
| **Primary tumour site** | | |
| Colon | | 217 (51.4) |
| Rectum | | 144 (34.1) |
| Treatment modalities for rectal cancer | Surgery only | 27 (18.7) |
| | Chemotherapy OR radiotherapy only | 14 (9.7) |
| | Surgery + chemotherapy OR radiotherapy | 45 (31.3) |
| | Surgery + chemotherapy + radiotherapy | 31 (21.5) |
| | Chemotherapy + radiotherapy | 4 (2.7) |
| | None | 23 (16) |
| Colorectal | | 31 (7.3) |
| Recto sigmoid | | 30 (7.1) |
| **Histopathology** | | |
| Adenocarcinoma NOS | | 333 (78.9) |
| Mucinous adenocarcinoma | | 53 (12.6) |
| Signet ring cell adenocarcinoma | | 20 (4.7) |
| Tubulovillous adenocarcinoma | | 7 (1.7) |
| Other | | 5 (1.2) |
| Unspecified | | 4 (0.9) |
| **Tumour grade** | | |
| Well differentiated | | 198 (46.9) |
| Moderately differentiated | | 84 (19.9) |
| Poorly differentiated | | 30 (7.1) |
| Undifferentiated | | 4 (0.9) |
| Unspecified | | 106 (25.1) |
| **Size of the tumour** | | |
| TX | | 12 (2.8) |
| T1 | | 7 (1.7) |
| T2 | | 62 (14.7) |
| T3 | | 197 (46.7) |
| T4 | | 113 (26.8) |
| Unspecified | | 31 (7.3) |
| **Lymph node metastasis** | | |
| Node negative | | 79 (18.7) |
| Node positive | | 102 (24.2) |
| Node unknown | | 241 (57.1) |
| **Distant metastasis** | | |
| MX | | 37 (8.8) |
| MO | | 230 (54.5) |
| M1 | | 143 (33.9) |
| Unspecified | | 12 (2.8) |
| **Lymph vascular invasion** | | |
| Yes | | 6 (1.4) |
| No | | 38 (9.0) |
| Unspecified | | 378 (89.6) |
| **Perineural invasion** | | |
| Yes | | 2 (0.5) |

(*Continued*)

**Table 2.** (Continued)

| Variables | n (%) |
|---|---|
| No | 2 (0.5) |
| Unspecified | 418 (99.1) |
| **Pre-operative CEA level** | |
| Not elevated ($\leq$5ng/ml) | 16 (3.8) |
| Elevated (>5ng/ml) | 14 (3.3) |
| Unspecified | 392 (92.9) |
| **Treatment modalities** | |
| Surgery only | 98 (23.2) |
| Chemotherapy OR radiotherapy only | 25 (5.9) |
| Surgery + chemotherapy OR radiotherapy | 208 (49.3) |
| Surgery + chemotherapy + radiotherapy | 49 (11.6) |
| Chemotherapy + radiotherapy | 6 (1.4) |
| None | 36 (8.5) |

NOS = not otherwise specified, CEA = carcinoembryonic Antigen.

who did not receive any cancer treatment during the follow-up time in TASH. This is because patients with advanced disease are not fit for chemotherapy or radiotherapy and refusal to initiate treatment. Enough data was not obtained for variables like pre-operative CEA level, lymphovascular and perineural invasion. Hence, not data exploration was performed for these variables.

### Result of a multivariable Cox regression model

To identify prognostic factors, variables with p-value < = 0.25 at the univariate analysis (Tables 3 and 4) were considered for the multiple Cox PH regression model. Hence, age at diagnosis (p = 0.159), comorbidity (p = 0.100), primary tumor site (p<0.01), histopathology of CRC (p = 0.011), tumor grade (p<0.001), size of the tumor (p<0.001), lymph node metastasis (p<0.001), distant metastasis (<0.001), treatment modality (p<0.001) and sex (due to its

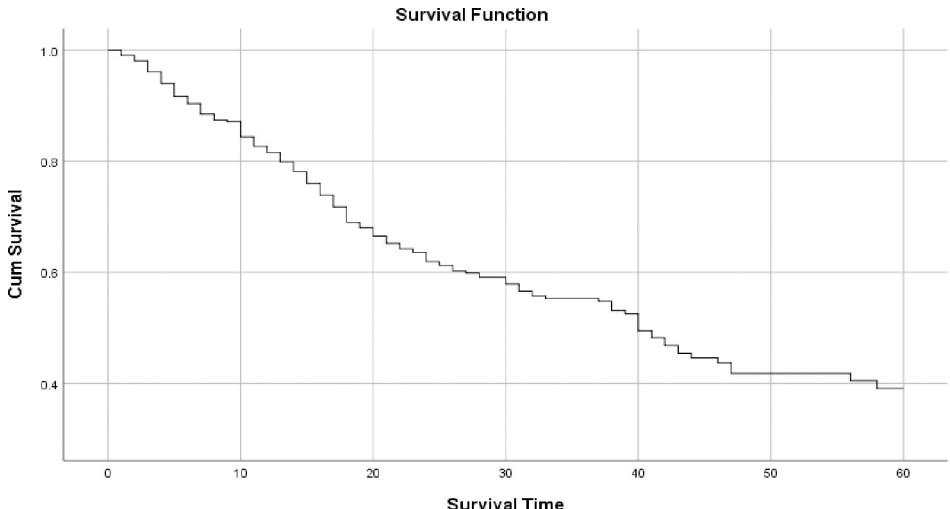

**Fig 1. Overall survival probability of colorectal cancer patients in TASH, Addis Ababa, Ethiopia, 2012–2016.**

**Table 3. Median survival, survival rate and log-rank test for univariate analysis of socio-demographic characteristics and comorbidities.**

| Variable | n | Median survival in month | Five-year survival rate (%) | Log-rank test | |
|---|---|---|---|---|---|
| | | | | $x^2$(df) | P-value |
| **Age** | | | | 7.96(5) | 0.159 |
| ≤29 | 50 | 23 | 29 | | |
| 30–39 | 79 | 39 | 45 | | |
| 40–49 | 94 | 43 | 35 | | |
| 50–59 | 91 | 40 | 32 | | |
| 60–69 | 77 | 60 | 50 | | |
| ≥70 | 31 | 25 | 7 | | |
| **Sex** | | | | 5.82(1) | 0.468 |
| Male | 262 | 39 | 33 | | |
| Female | 160 | 46 | 34 | | |
| **Place of residence** | | | | 0.26(1) | 0.871 |
| Addis Ababa | 187 | 39 | 36 | | |
| Out of Addis Ababa | 235 | 40 | 29 | | |
| **Comorbidities** | | | | 2.70(1) | 0.100 |
| Yes | 52 | 21 | 23 | | |
| No | 370 | 41 | 35 | | |
| **HIV/AIDS** | | | | 2.09(1) | 0.148 |
| Yes | 7 | 13 | 29 | | |
| No | 415 | 40 | 33 | | |
| **Diabetic mellitus** | | | | 0.74(2) | 0.964 |
| Yes | 15 | 60 | 26 | | |
| No | 406 | 40 | 35 | | |
| **Hypertension** | | | | 0.50(1) | 0.477 |
| Yes | 28 | 60 | 60 | | |
| No | 394 | 39 | 31 | | |

$x^2$ = **chi square**, DF = degree of freedom, HIV/AIDS = human immune virus/acquired immunodeficiency syndrome.

importance in oncology study) were entered to the final multiple Cox regression model. The proportional hazard assumption was tested using scaled-Schoenfeld residuals and the global test for the model (p = 0.0584). We failed to reject the null hypothesis that the hazards are proportional. The scaled Schoenfeld residuals were also plotted for each variable and the slope was closed to zero for variables in the model and Kaplan Meier curves were parallel for each category of variables versus time. These imply that the proportional hazard assumption was not violated for this data. Results of the final multivariable Cox regression analysis are presented in Table 5. From greatest to least powerful: distant metastasis (P <0.001), lymph node involvement (P <0.01), treatment modalities (p< 0.01) and primary tumour site (P <0.05) were significantly associated with survival rate.

Patients diagnosed with rectal cancer had 76% (HR 1.761, 95% CI: 1.173–2.644) increased risk to die compared to colon cancer. Node positive patients were 3.146 (95% CI: 1.629–6.078) times likely to die compared to node-negative patients. The risk of death for metastatic cancer was 4.221 (95% CI: 2.788–6.392) times to non-metastatic patients. For treatment modality, risk of death was 36.1% lesser (HR: 0.639 (95% CI: 0.418–0.977)) and 47.9% lesser (HR: 0.521 (95% CI: 0.279–0.973)) in those patients who received adjuvant chemotherapy or adjuvant radiotherapy and who received adjuvant chemotherapy and adjuvant radiotherapy, respectively compared to patients who had only surgical resection (Table 5).

**Table 4. Median survival, survival rate and log-rank test for univariate analysis of pathological and clinical characteristics.**

| Variable | n | Median survival in month | Five-year survival rate (%) | Log-rank test | |
|---|---|---|---|---|---|
| | | | | $x^2$(df) | P-value |
| **Primary tumour site** | | | | 17.00(3) | **0.001** |
| Colon | 217 | 60 | 53 | | |
| Rectum | 144 | 25 | 16 | | |
| Colorectal | 31 | 57 | 21 | | |
| Recto sigmoid | 30 | 41 | 27 | | |
| **Histopathology** | | | | 14.93(5) | **0.011** |
| Adenocarcinoma NOS | 333 | 42 | 36 | | |
| Mucinous adenocarcinoma | 53 | 24 | 15 | | |
| Signet ring cell adenocarcinoma | 20 | 23 | - | | |
| Tubulovillous adenocarcinoma | 7 | 5 | - | | |
| **Tumour grade** | | | | 52.63(4) | **0.0001** |
| Well differentiated | 198 | 60 | 57 | | |
| Moderately differentiated | 84 | 41 | 31 | | |
| Poorly differentiated | 30 | 15 | 41 | | |
| Undifferentiated | 4 | 40 | - | | |
| Unspecified | 106 | 18 | 6 | | |
| **Size of the tumour** | | | | 55.46(5) | **0.0001** |
| TX | 12 | 12 | - | | |
| T1 | 7 | 52 | - | | |
| T2 | 62 | 60 | 68 | | |
| T3 | 197 | 46 | 38 | | |
| T4 | 113 | 21 | 22 | | |
| Unspecified | 31 | 15 | - | | |
| **Lymph node metastasis** | | | | 34.584(2) | **0.0001** |
| Node negative | 79 | 60 | 77 | | |
| Node positive | 102 | 24 | 14 | | |
| Node unknown | 241 | 32 | 28 | | |
| **Distant metastasis** | | | | 142.60(3) | **0.0001** |
| MX | 37 | 30 | 17 | | |
| M0 | 230 | 60 | 55 | | |
| M1 | 143 | 14 | 2 | | |
| **Treatment modalities** | | | | 66.09(5) | **0.0001** |
| Surgery only | 98 | 39 | 25 | | |
| Surgery OR radiotherapy only | 25 | 14 | - | | |
| Surgery + chemotherapy OR radiotherapy | 208 | 46 | 43 | | |
| Surgery + chemotherapy + radiotherapy | 49 | 73 | 38 | | |
| Chemotherapy + radiotherapy | 6 | 12 | - | | |
| None | 36 | 6 | 14 | | |

$x^2$ = **chi square**, DF = degree of freedom, NOS = not otherwise specified.

## Discussion

This study provided estimates of survival time and identified prognostic factors of colorectal cancer patients in the only oncology referral Centre in Ethiopia over the period 2012–2016. Our findings revealed that the median age at diagnosis was 46 and 52.8% of those diagnosed were under 50, which shows the majority of colorectal cancer patients in Ethiopia are young

**Table 5. Multivariable survival analysis of prognostic factors in colorectal cancer patients.**

| Prognostic variables | Estimate | Standard error | p-value | HR | CI (95%) |
|---|---|---|---|---|---|
| **Age** | | | 0.263 | | |
| ≤29 | | | | 1 | |
| 30–39 | -0.266 | 0.329 | 0.419 | 0.767 | 0.403–1.460 |
| 40–49 | -0.170 | 0.327 | 0.582 | 0.836 | 0.442–1.582 |
| 50–59 | -0.067 | 0.300 | 0.822 | 0.935 | 0.520–1.682 |
| 60–69 | 0.046 | 0.325 | 0.908 | 1.038 | 0.550–1.958 |
| ≥70 | -0.740 | 0.376 | 0.052 | 0.484 | 0.233–1.006 |
| **Sex** | | | | | |
| Male | 0.291 | 0.175 | 0.097 | 1.338 | 0.949–1.887 |
| Female | | | | 1 | |
| **Comorbidities** | | | | | |
| Yes | 0.146 | 0.239 | 0.543 | 1.157 | 0.724–1.848 |
| No | | | | 1 | |
| **Primary tumour site** | | | **0.036** | | |
| Colon | | | | 1 | |
| Rectum | 0.566 | 0.207 | 0.006 | 1.761 | 1.173–2.644 |
| Colorectal | -0.023 | 0.355 | 0.948 | 0.977 | 0.488–1.959 |
| Recto sigmoid | 0.446 | 0.339 | 0.188 | 1.562 | 0.804–3.035 |
| **Histopathology** | | | 0.249 | | |
| Adenocarcinoma NOS | | | | 1 | |
| Mucinous adenocarcinoma | 0.373 | 0.245 | 0.128 | 1.452 | 0.899–2.346 |
| Signet ring cell adenocarcinoma | 0.487 | 0.388 | 0.210 | 1.628 | 0.760–3.485 |
| Tubulovillous adenocarcinoma | 0.147 | 0.522 | 0.778 | 1.159 | 0.417–3.223 |
| **Tumour grade** | | | 1.180 | | |
| Well-differentiated | | | | 1 | |
| Moderately differentiated | 0.311 | 0.221 | 0.159 | 1.364 | 0.885–2.102 |
| Poorly differentiated | 0.579 | 0.325 | 0.066 | 1.817 | 0.060–3.439 |
| Undifferentiated | 1.220 | 1.055 | 0.247 | 3.389 | 0.429–26.786 |
| **Size of the tumour** | | | 0.051 | | |
| T1 | 0.112 | 0.628 | 8.859 | 1.118 | 0.326–3.832 |
| T2 | -0.336 | 0.323 | 0.297 | 0.714 | 0.380–1.344 |
| T3 | -0.195 | 0.199 | 0.326 | 0.823 | 0.557–1.216 |
| T4 | | | | 1 | |
| **Lymph node metastasis** | | | **0.003** | | |
| Node negative | | | | 1 | |
| Node positive | 1.146 | 0.336 | 0.001 | 3.146 | 1.629–6.078 |
| **Distant metastasis** | | | **0.000** | | |
| M0 | | | | 1 | |
| M1 | 1.440 | 0.212 | 0.000 | 4.221 | 2.788–6.392 |
| **Treatment modalities** | | | **0.007** | | |
| Surgery only | | | | 1 | |
| Surgery OR radiotherapy only | -0.051 | 0.334 | 0.879 | 0.950 | 0.494–1.828 |
| Surgery + chemotherapy OR radiotherapy | -0.447 | 0.217 | 0.039 | 0.639 | 0.418–0.977 |
| Surgery + chemotherapy + radiotherapy | -0.653 | 0.319 | 0.041 | 0.521 | 0.279–0.973 |
| Chemotherapy + radiotherapy | 0.142 | 0.572 | 0.803 | 1.153 | 0.376–3.537 |
| None | 0.575 | 0.350 | 0.100 | 1.778 | 0.895–3.531 |

HR = hazard ratio, CI = confidence interval, NOS = not otherwise specified.

adults and middle-aged adults whereas the median age at diagnosis was 70 in Germany. Data from Oman showed that the median age was 56 years and a majority of the patients, 61.1%, were older than 50 years. Similarly, in Jordan, the median age was 62 for males and 58 for females. More than half of the patients were above age 60 years, which differs from our study result. The age structure of Ethiopia could be a contributing factor. It might also relate to the unhealthy nutritional habit in Ethiopia (high intake of fats and red meats) and lack of birth registration may contribute as most patients from the rural area might not know their exact age [21–23].

Around 34% of the patients had metastatic cancer at presentation. This is similar to a report from Oman (36.8) and Kenya (29.1%) but in contrast with reports from Germany (28.1%), in Ghana (24.9%), in Malaysia (24.3%), in Jordan (17.2%) and Iran (6.8%) of patients were meta-static [13,16,21–25]. The late presentation might be due to patients' low awareness of colorectal cancer sign and symptoms, lack of screening program, lengthy and poor referral system in Ethiopia. This might have contributed to the observed low survival of patients treated in Tikur Anbesa Specialized Hospital in Addis Ababa, Ethiopia.

In our current study, the five-year overall survival (OS) rate was 33%, much lower than the OS rate in developed countries. Highest five-year survival rate (72%) are in Israel and North Korea. Survival varies from 65% to 70% in North America, Europe and Australia, 63% in Germany and 55% in other developed countries [17,18,21]. This could be primarily due to increase in colorectal cancer screening, removal of precancerous adenomas, timely and advanced treatment in developed countries. In developing Asian countries, the five-year survival rate of CRC patients ranged from 33% to 56.9% in Iran, 58.2% in Jordan, 43% & 48.7% in Malaysia, 42% in Oman, 39% in India, 38.6% in Thailand which are above the survival rate of patients in Ethiopia [13–15,18,22–26]. The reason for this difference could be lack of screening programs to detect cancer at an early stage, poorly developed infrastructure of cancer health cares in Ethiopia. Treatments are also very limited and inaccessible especially for patients living far from TASH, the only oncology and cancer referral centre in the country. The five years overall survival rates of colorectal cancer patients in a teaching hospital in Ghana (n = 221) was 16% which is below the overall survival rate in this study [16]. Variation in methodology, population characteristics and observed number of events may have contributed to these differences in findings.

In our study, age at diagnosis was not a significant prognostic factor affecting survival. This is consistent with studies conducted in Malaysia and Ghana [16,25]. However, it was inconsistent with studies conducted in India, Iran, Jordan and Thailand [13,14,23,27,28]. This might be because the majority of the study participant was young compared to patients in those studies. Comorbidity is not also a significant prognostic factor affecting patient's survival. This finding is also in line with reports from Ghana, Iran and Malaysia [15,16,25].

In multivariable Cox regression model, primary tumor site (P <0.05), lymph node involvement (P <0.01), distant metastasis (P <0.001) and treatment modalities (p< 0.01) were significant prognostic factors for colorectal cancer patients survival. Patients diagnosed with rectal cancer had 76% (HR = 1.761) increased hazard to death compared to colon cancer patients. This is in agreement with reports from Iran where rectal cancer had a high risk of death and determines prognosis significantly [13]. In our study, it is associated with inadequacy of treatment.

In this study, the hazard of death was significantly higher for those patients who were diagnosed with a positive lymph node with (HR = 3.146) compared to node-negative. This finding is in line with reports from Ghana, Malaysia, Thailand and Iran [13,14,16,25]. The other finding of this study is the risk of death was four-fold higher (HR = 4.221) for those who presented with metastatic cancer compared to non-metastatic cancer patients. This finding is comparable

to those reports found in Ghana, Thailand, Jordan, Malaysia and Iran [13,14,16,23,25]. Node involvement and metastatic cancer at presentation were the major prognostic factors for colorectal cancer patients.

The finding of this study demonstrates that the risk of death was significantly lesser (HR = 0.639 &0.521) among those who received adjuvant chemotherapy or/and adjuvant radiotherapy compared to those who only received surgical resection. This is similar to reports from Ghana, Thailand, Iran and Malaysia that adjuvant therapy was a good prognostic factor [13,14,16,25].

The study was conducted in the only oncology referral centre in the country that provides services for the majority of cancer cases from all over the country. Hence, our findings may reflect the situation in the whole of Ethiopia. As the data is from a secondary source, some prognostic factors such as family history, lymph node involvement, lymph vascular invasion, perineural invasion and pre-operative CEA level were missing from medical records during data collection and cleansing. Also, important information on diet and physical activity, alcohol intake, smoking history, body mass index and complication after treatment were not included in our study. The other limitation is that only seven CRC patients died in the institution with a report of detailed causes of death. The death of the remaining 168 patients and causes of deaths were ascertained by telephone interview, not from death report which may lead to over detection of death as a result of cause ascertainment bias.

## Conclusion

This study estimates a low overall five-year survival rate. Majority of the patient's s were young adults and diagnosed with advanced cancer stage. Patient's survival is largely influenced by advanced stage at presentation and delayed in the administration of adjuvant therapy. Primary tumour site, lymph node involvement, distant metastasis and treatment modalities were important prognostic factors for colorectal cancer. Rectal cancer, node-positive, and metastatic cancer at a presentation in colorectal cancer patients decreased the survival rate. Whereas, adjuvant therapy increased survival rate. Administration of adjuvant chemotherapy through improving access to chemotherapy and radiotherapy will improve outcome.

## Supporting information

**S1 Data.**
(SAV)

## Acknowledgments

The authors would like to thank the TASH radiotherapy centre for granting permission to extract the data for this study and Mr. Solomon Asmare, for his support throughout this study and workers at the card room.

## Author Contributions

**Conceptualization:** Eyob Kebede Etissa, Birhanu Teshome Ayele.

**Data curation:** Eyob Kebede Etissa.

**Formal analysis:** Eyob Kebede Etissa.

**Methodology:** Eyob Kebede Etissa.

**Resources:** Eyob Kebede Etissa.

**Software:** Eyob Kebede Etissa.

**Supervision:** Birhanu Teshome Ayele.

**Writing – original draft:** Eyob Kebede Etissa.

**Writing – review & editing:** Eyob Kebede Etissa, Mathewos Assefa, Birhanu Teshome Ayele.

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
