## [Decision Letter · Decision Letter 0]

25 Sep 2020

PONE-D-20-03951

Prognosis of Colorectal Cancer in Tikur Anbessa Specialized Hospital, the Only Oncology Center in Ethiopia.

PLOS ONE

Dear Dr. Etissa,

Thank you for submitting your manuscript to PLOS ONE. After careful consideration, we feel that it has merit but does not fully meet PLOS ONE’s publication criteria as it currently stands. Therefore, we invite you to submit a revised version of the manuscript that addresses the points raised during the review process.

We look forward to receiving your revised manuscript.

Kind regards,

Ludmila Vodickova, M.D., PhD

Academic Editor

PLOS ONE

Journal Requirements:

3. Thank you for clarifying that the participants provided verbal informed consent for the phone interviews conducted in this study. At this time, we also ask that you provide additional information about the consent sought to access the medical records of the patients. In the ethics statement in the Methods and online submission information, please ensure that you have specified (1) whether consent was informed and (2) what type you obtained (for instance, written or verbal, and if verbal, how it was documented and witnessed). If your study included minors, state whether you obtained consent from parents or guardians. If the need for consent was waived by the ethics committee, please include this information.

4. Please provide a sample size and power calculation in the Methods, or discuss the reasons for not performing one before study initiation. 

Reviewers' comments:

Reviewer's Responses to Questions

**Comments to the Author**

1. Is the manuscript technically sound, and do the data support the conclusions?

Reviewer #1: Partly

Reviewer #2: Yes

Reviewer #3: No

2. Has the statistical analysis been performed appropriately and rigorously? 

Reviewer #1: I Don't Know

Reviewer #2: Yes

Reviewer #3: No

3. Have the authors made all data underlying the findings in their manuscript fully available?

Reviewer #1: Yes

Reviewer #2: Yes

Reviewer #3: Yes

4. Is the manuscript presented in an intelligible fashion and written in standard English?

Reviewer #1: Yes

Reviewer #2: Yes

Reviewer #3: No

5. Review Comments to the Author

Reviewer #1: I have not enough knowledge to evaluate easily of the manucript conclusions. Authors have not know the lymph node status of more than half of the participants probably due to surgical or pathological problems. We can not predict exactly the prognosis of patients if we do not know the true stating . Also, there is no any knowledge about treatment options of the treated patients with chemotherapy in the text.

Reviewer #2: The work is generally interesting with the use of research methods available to the authors (limitation - telephone interview and incomplete documentation) and correct statistical methods. The authors are aware of the limitations that mainly result from the lack of data, but they are not exposed too much. However, the work requires editing.

Introduction.

- Anatomically, the large intestine consists of the colon and rectum,

- repeat: cancer stage is the size of the tumor, the condition of the lymph nodes and the presence / absence of metastases to distant organs,

- prognostic factors are additionally: tumor maturity, complicated colorectal cancer (perforation, obstruction, bleeding), treatment method, complications (morbidity) after treatment, e.g. local infection after surgery, high body temperature; factor not related to tumor and patient: the surgeon's learning curve.

- the prognostic factors and the probability of survival should be written in order (abstract, introduction)

Material and methods.

- the 3rd sentence can be moved to the statistics part,

- telephone interview not very precise, in addition, data from official (state) sources should be obtained,

- How many patients did not meet the inclusion criteria

Results

- the location of the cancer in individual parts of the colon (caecum, ascending, transverse, descending, sigmoid colon) should be indicated,

- almost 60% of patients did not have a specific cancer stage, so it is difficult to define indications for adjuvant treatment,

- in a multivariate analysis, the prognostic factors related to the tumor should be separated from the prognostic factors related to the patient, e.g. comorbidities, age, sex. In the results section - did not specify clearly which of the prognostic factors in the multivariate analysis had the greatest prognostic power; the factors should be systematized from having the greatest bargaining power to the least powerful.

Discussion

- not all research results were discussed in the discussion section. The limitations of the study were not discussed in detail. which mainly occurred due to lack of data.

Conclusions

- should be short and concise, be based on work and correspond to predetermined endpoints

- Not all content in the conclusions results from work.

Table 1 Complete the title with medical history data in accordance with the table

Fig. A, B titles: probability of survival

Fig. B - it is better to present the probability of survival in individual stages, only stage IV survival is presented here. This table can be omitted.

Reviewer #3: Overall credible registry based study with a statistically sound sample size.

Unfortunately several major errors were noted:

- It appears that the authors evaluate the role of adjuvant therapy as an important prognostic factor, however the analysis was done on a very heterogeneous subset of patients. Ideally the role and utility of adjuvant therapy and possible survival benefit should be limited to patients in Stage 3 colon cancer where a documented survival benefit has been established based on influential trials like the MOSAIC trial.

- It is unclear if all patients who underwent adjuvant therapy had surgery. It makes intuitive sense that they did; however this was not specified.

- The authors report that 57% of patients had unknown LN status. This is a very large patient cohort with unclear LN status. Hence the decision making process for adjuvant therapy is questionable given that lymph nodal status is a key prognostic variable for OS/Adjuvant therapy. This raises several questions as to the validity of the analysis given that surgery/adjuvant therapy did not conform to current accepted standards of care.

- For the rectal cancer subset, the authors report an increased risk of death. However, it is unclear if patents received any neoadjuvant/adjuvant therapy and surgery/type of surgery LAR/APR. Any inadequately treated cancer is associated with an increased risk of death.

- In the conclusion section, authors report incorporation of screening and early detection are crucial to "increasing survival time". Screening and early detection were not included in this analysis; hence this is an inappropriate conclusion to make within the context of this current research.

- Several English and grammatical errors were also noted through the manuscript

6. PLOS authors have the option to publish the peer review history of their article (what does this mean?). If published, this will include your full peer review and any attached files.

Reviewer #1: No

Reviewer #2: **Yes: **Andrzej Nowicki

Reviewer #3: No

---

## [Author Response · Author response to Decision Letter 0]

12 Nov 2020

Comments from an academic editor

Response: The correction has been made to meet the requirements.

2. We suggest you thoroughly copyedit your manuscript for language usage, spelling, and grammar. If you do not know anyone who can help you do this, you may wish to consider employing a professional scientific editing service.  Whilst you may use any professional scientific editing service of your choice, PLOS has partnered with both American Journal Experts (AJE) and Editage to provide discounted services to PLOS authors. Both organizations have experience helping authors meet PLOS guidelines and can provide language editing, translation, manuscript formatting, and figure formatting to ensure your manuscript meets our submission guidelines. To take advantage of our partnership with AJE, visit the AJE website (http://learn.aje.com/plos/) for a 15% discount off AJE services. To take advantage of our partnership with Editage, visit the Editage website (www.editage.com) and enter referral code PLOSEDIT for a 15% discount off Editage services.  If the PLOS editorial team finds any language issues in text that either AJE or Editage has edited, the service provider will re-edit the text for free.

Response: All language usage, spelling, and grammar errors pointed out by the academic editor and reviewers were corrected throughout the manuscript.

3. Thank you for clarifying that the participants provided verbal informed consent for the phone interviews conducted in this study. At this time, we also ask that you provide additional information about the consent sought to access the medical records of the patients. In the ethics statement in the Methods and online submission information, please ensure that you have specified (1) whether consent was informed and (2) what type you obtained (for instance, written or verbal, and if verbal, how it was documented and witnessed). If your study included minors, state whether you obtained consent from parents or guardians. If the need for consent was waived by the ethics committee, please include this information.

Response: Thank you for pointing this out. We have provided additional information sought to access the medical records of the patients and how oral consent was witnessed and documented in the revised manuscript as: - 

‘’ To access the medical records ethical approval letter for the proposed study was obtained from GAMBY Medical and Business Sciences College institutional research ethics review committee (GCA.A 20/2010) to Tikur Anbessa Specialized Hospital. Permission letter was obtained from Tikur Anbessa Specialized Hospital to extract data from medical records. Written consent could not be obtained as the patients were not at the facility. Verbal informed consent was obtained for patient’s eligible for the phone interview, witnessed by two experienced data collectors who are not a part of the study and documented via written note. Minors were not included in the study.’’

4. Please provide a sample size and power calculation in the Methods, or discuss the reasons for not performing one before study initiation.

Response: As requested by the editor, sample size consideration is provided. The paragraph now reads:

‘’ Sample size determination started by determining d (number of events) because of the number of observed events matters beyond the total number of patients in survival data analysis. The required total number of events (Freedman 1982) was calculated using:

d =

That Δ=hazard ratio is constant in time. For the two-sided log-rank test (H0: Δ = 1 vs HA: Δ ≠ 1).

Then, the required total number of patients can be calculated as A previous similar study conducted in Ghana showed that s (t) = 0.16, thus we used. We also know that.

To obtain the total sample size required to yield a power of 90%, a hazard ratio of 1.832 and 5% two-sided level of significance, the required number of events is:

=

Accordingly, 

Adjusting for a loss of 20%. .

The required sample size was 172. However, all colorectal cancer patients treated in TASH from 2012 to 2016 and who fulfil the inclusion criteria were included in the study. Hence the sample used for this study was 422. ’’

Response: Thank you for pointing this out, we have updated data availability statement.

 All relevant data are within the manuscript and its Supporting Information files. Anonymous data (uploaded as *data* file)

6. PLOS authors have the option to publish the peer review history of their article (what does this mean?). If published, this will include your full peer review and any attached files.

Response: No

Reviewer #1

Comment: I have not enough knowledge to evaluate easily of the manuscript conclusions. Authors have not known the lymph node status of more than half of the participants probably due to surgical or pathological problems. We cannot predict exactly the prognosis of patients if we do not know the true staging. Also, there is no any knowledge about treatment options of the treated patients with chemotherapy in the text.

Response: We agree with the reviewer assessment. As you noted more than half (57.1%) study participant’s lymph node status was not known. To differentiate Stage II from stage III at least 10-12 lymph node should be submitted and examined by a pathologist – meaning we do not know the staging in this group of patients. Therefore, TNM staging was not considered as a prognostic factor, as it could not predict the exact prognosis of patients. Then, we made a prediction based on the size and location of tumor, lymph node metastasis (node-positive versus node-negative) and distant metastasis (Metastatic versus non-metastatic colorectal cancer). The therapeutic option was exclusively based on 5 – Fluorouracil (5-FU) for almost all cases as standard care of treatment. That’s why chemotherapy treatment option information not provided.

Reviewer #2

Comment: The work is generally interesting with the use of research methods available to the authors (limitation - telephone interview and incomplete documentation) and correct statistical methods. The authors are aware of the limitations that mainly result from the lack of data, but they are not exposed too much. However, the work requires editing.

Response: Thank you!

Introduction

Comment: Anatomically, the large intestine consists of the colon and rectum,

Response: Thank you. We have amended the text: Page 4, line 69

‘‘Colorectal cancer is a cancer of the large intestine. Anatomically, it also is known as colon cancer or rectal cancer but when both present with similar features they are termed as colorectal cancer (CRC).’’

Comment: repeat: cancer stage is the size of the tumor, the condition of the lymph nodes and the presence / absence of metastases to distant organs,

Response: The text has been revised as suggested:

 ‘‘Various studies have shown that CRC survival is mainly determined by age at diagnosis, complicated colorectal cancer (perforation, obstruction, bleeding), tumour site, tumour grade, tumour maturity, tumour size and location, lymph node involvement, distant metastasis, treatment modalities and complication after treatment. Some literature reported that sex, educational status, family history, pre-operative carcinoembryonic Antigen (CEA) level and lymphovascular invasion as significant prognostic factors.’’

Comment: Prognostic factors are additionally: tumor maturity, complicated colorectal cancer (perforation, obstruction, bleeding), treatment method, complications (morbidity) after treatment, e.g. local infection after surgery, high body temperature; factor not related to tumor and patient: the surgeon's learning curve.

Response: The suggested correction has been made except factor not related to tumor and patient: the surgeon’s learning curve. We couldn’t obtain evidence that shows surgeon’s learning curve affecting the survival of colorectal cancer: 

‘‘Various studies have shown that CRC survival is mainly determined by age at diagnosis, complicated colorectal cancer (perforation, obstruction, bleeding), tumour site, tumour grade, tumour maturity, tumour size and location, lymph node involvement, distant metastasis, treatment modalities and complication after treatment. Some literature reported that sex, educational status, family history, pre-operative carcinoembryonic Antigen (CEA) level and lymphovascular invasion as significant prognostic factors.’’

References

Zare-Bandamiri M, Khanjani N, Jahani Y, Mohammadianpanah M. Factors Affecting Survival in Patients with Colorectal Cancer in Shiraz, Iran. Asian Pacific Journal of Cancer Prevention. 2016;17(1):159-63.

Agyemang-Yeboah F, Yorke J, Obirikorang C, Batu E, Acheampong E, Frimpong E, et al. Colorectal cancer survival rates in Ghana: A retrospective hospital-based study. PLOS ONE. 2018.

Rasouli M, Moradi G, Roshani D, Nikkhoo B, Ghaderi E, Ghaytasi B. Prognostic factors and survival of colorectal cancer in Kurdistan province, Iran: A population-based study (2009-2014). Medicine (Baltimore). 2017;96(6).

Laohavinij S, Maneechavakajorn J, P T. Prognostic factors for survival in colorectal cancer patients. J Med Assoc Thailand. 2010;93(10):1156-66.

Comment: The prognostic factors and the probability of survival should be written in order (abstract, introduction)

Response: The suggested correction have been made in both sections.

Materials and methods

Comment: The 3rd sentence can be moved to the statistics part,

Response: Thanks for the suggestion. We moved the 3rd sentence to the statistics part and changes the statistical analysis part to sample size consideration and analysis plan. 

Comment: Telephone interview not very precise, in addition, data from official (state) sources should be obtained,

Response: We fully agree with our reviewer. A telephone interview is not the ideal way of obtaining such information. Unfortunately in Ethiopia, as it is common in most Low and Middle-Income Countries (LMIC), there is no organized vital registry system or any other official source to obtain the required information. Hence, we conducted a phone interview with this in mind. We believe that the telephone interview has helped us to ascertain the vital status of some study participants though not the precise way to obtain the right information.

Comment: How many patients did not meet the inclusion criteria?

Response: ‘‘Thirteen patients not meeting the exclusion criteria were excluded from the study.’’ We have added the sentence in the manuscript. Page 6, line 115

Results

Comment: The location of the cancer in individual parts of the colon (caecum, ascending, transverse, descending, and sigmoid colon) should be indicated,

Response: We thank the reviewer for pointing out that. We did not indicate the location of cancer in individual parts of the colon. The problem is that more than 90% of the study participants were referred from other institutions. Majority referral papers do not clearly state the exact part of the affected colon, simply put the diagnosis as colon cancer or it is lost from the medical record. This was noticed while pre-testing the data extraction tool.

Comment: Almost 60% of patients did not have a specific cancer stage, so it is difficult to define indications for adjuvant treatment,

Response: Thanks, you have raised an important point here. So if the nodal status is not known patients will be treated with adjuvant chemotherapy as high-risk stage II colon cancer depending on presence or absence of high-risk features (localized perforation, lymph vascular invasion, perineural invasion, bowel obstruction ….) despite its use in stage II is controversial (small but statistically significant benefit), but if the lymph nodes are reported and negative adjuvant chemotherapy is avoided.

Comment: In a multivariate analysis, the prognostic factors related to the tumor should be separated from the prognostic factors related to the patient, e.g. comorbidities, age, sex. 

Response: We respectfully disagree with the reviewer comment. Prognostic factors related to the patient and tumor must be analyzed together in multivariate analysis. This is due to patient-related factors (comorbidities, age, and sex) have a direct impact in tumor. For example, comorbidities may result from an opportunity for screening and early diagnosis or delay in diagnosis, so should be analyzed together not to miss important prognostic factors that are significant when analyzed together but insignificant when separately analyzed.

Comment: In the results section - did not specify clearly which of the prognostic factors in the multivariate analysis had the greatest prognostic power; the factors should be systematized from having the greatest bargaining power to the least powerful. 

Response: The correction has been made. The sentence now reads: page 17, line 267-270

‘‘Results of the final multivariable Cox regression analysis are presented in Table 5. From greatest to least powerful: distant metastasis (P <0.001), lymph node involvement (P <0.01), treatment modalities (p< 0.01) and Primary tumour site (P <0.05) were significantly associated with survival rate.’’

Discussion

Comment: Not all research results were discussed in the discussion section. The limitations of the study were not discussed in detail. Which mainly occurred due to lack of data.

Response: Agree, not all research results were discussed. We focused on major study findings, which calls for interpretation. As suggested by the reviewer, we have amended the limitation of the study occurred due to lack of data:

‘‘As the data is from a secondary source, some prognostic factors such as family history, lymph node involvement, lymph vascular invasion, perineural invasion and pre-operative CEA level were missing from medical records during data collection and cleansing. Also, important information on diet and physical activity, alcohol intake, smoking history, body mass index and complication after treatment were not included in our study.’’

Conclusion

Comment: should be short and concise, be based on work and correspond to predetermined endpoints

Not all content in the conclusions results from work.

Response: By the reviewer’s comment, we have made corrections.

‘‘This study estimates a low overall five-year survival rate. Majority of the patient’s s were young adults and diagnosed with advanced cancer stage. Patient’s survival is largely influenced by advanced stage at presentation and delayed in the administration of adjuvant therapy. Primary tumour site, lymph node involvement, distant metastasis and treatment modalities were important prognostic factors for colorectal cancer. Rectal cancer, node-positive, and metastatic cancer at a presentation in colorectal cancer patients decreased the survival rate. Whereas, adjuvant therapy increased survival rate. Administration of adjuvant chemotherapy through improving access to chemotherapy and radiotherapy will improve outcome.” 

Comment: Table 1 Complete the title with medical history data in accordance with the table

Response: The Table title is modified as follows: 

Table 1 title ‘‘Table 1: Socio-demographic characteristics and co-morbidity history of colorectal cancer patients (n=422) in Tikur Anbesa Specialized Hospital (TASH), Addis Ababa, Ethiopia, 2012-2016’’

Comment: Fig. A, B titles: probability of survival

Response: Thank you for this suggestion. Amended accordingly. ‘‘Figure 1: Overall survival probability of colorectal cancer patients in TASH, Addis Ababa, Ethiopia, 2012-2016’’ 

Comment: Fig. B - it is better to present the probability of survival in individual stages, only stage IV survival is presented here. This table can be omitted.

Response: As suggested by the reviewer, Fig B is omitted. In our data, more than half (57.1%) study participant’s lymph node status was unknown. To differentiate Stage II from stage III at least 10-12 lymph node should be submitted and examined by a pathologist – meaning we do not know the staging in this group of patients. We could not provide the probability of survival in an individual stage.

Reviewer #3

Comment: Overall credible registry based study with a statistically sound sample size.

Unfortunately several major errors were noted:

Thank you for your thorough review.

Comment: It appears that the authors evaluate the role of adjuvant therapy as an important prognostic factor, however the analysis was done on a very heterogeneous subset of patients. Ideally the role and utility of adjuvant therapy and possible survival benefit should be limited to patients in Stage 3 colon cancer where a documented survival benefit has been established based on influential trials like the MOSAIC trial.

Response: You have raised an important point here. We respectfully disagree that survival benefit should be limited to patients in stage III. In fact, in stage II its benefit is controversial (small but statistically significant), with ongoing studies seeking to confirm which markers might identify patients who would benefit. 

References

NCCN Guidelines for patients: Colon Cancer. 2018

Al B. Benson III, Alan P. Venook, Mahmoud M. Al-Hawary, Lynette Cederquist, Yi-Jen Chen, Kristen K. Ciombor, et al. NCCN Guidelines® Insights. JNCCN—Journal of the National Comprehensive Cancer Network. April 2018;16(4).

A. C. Adjuvant Treatement of Colorectal Cancer. Gastrointestinal Cancer Research. 2008;2((4 suppl 2)):S42-S6.

Kannarkatt J, Joseph J, Kurniali PC, Al-Janadi A, . BH. Advuvant Chemotherapy for Stage II Colon Cancer: A Clinical Dilemma. Jornal of oncology practice. 2017;13(4):233-41.

Yang Y, Yang Y, Yang H, Wang F, Wang HH, Chen Q, et al. Adjuvant Chemotherapy for Stage II Colon Cancer: Who Really Needs It. Cancer Management and Research. 2017;10.

Comment: It is unclear if all patients who underwent adjuvant therapy had surgery. It makes intuitive sense that they did; however this was not specified.

Response: Yes, all patients who underwent adjuvant therapy had surgery, intuitively known without any rational process. We have added a sentence to make it clear to our readers.

Now it reads: Page 11, line 202

“Nearly half, (49.3%), of the patients, had received adjuvant chemotherapy or radiotherapy after surgery. All patients who underwent adjuvant therapy had surgery.’’

Comment: The authors report that 57% of patients had unknown LN status. This is a very large patient cohort with unclear LN status. Hence the decision making process for adjuvant therapy is questionable given that lymph nodal status is a key prognostic variable for OS/Adjuvant therapy. This raises several questions as to the validity of the analysis given that surgery/adjuvant therapy did not conform to current accepted standards of care.

Response: Thank you for your constructive review. Patients with unclear LN status will be treated with adjuvant chemotherapy as high-risk stage II colon cancer depending on presence or absence of high-risk features despite its use in stage II is controversial (small but statistically significant benefit), but if the lymph nodes are reported and negative adjuvant chemotherapy is avoided.

According to the National Comprehensive Cancer Network (NCCN) clinical practice guideline, adjuvant chemotherapy is recommended as an option for patients with stage II colon cancer that has high-risk features.

Comment: For the rectal cancer subset, the authors report an increased risk of death. However, it is unclear if patents received any neoadjuvant/adjuvant therapy and surgery/type of surgery LAR/APR. Any inadequately treated cancer is associated with an increased risk of death.

Response: Thanks for the valuable comment. They might be associated with an increased risk of death for rectal cancer patients. We have added this valuable content in the manuscript as follows: page 11, line 193-195 and page 21, line 328

‘’ Forty-five (31.3 %) rectal cancer patients had adjuvant chemotherapy or radiotherapy after surgery, 31 (21.5) had adjuvant chemotherapy and radiotherapy after surgery and 23 (16%) did not receive any treatment.

Patients diagnosed with rectal cancer had 76% (HR = 1.761) increased hazard to death compared to colon cancer patients. This is in agreement with reports from Iran where rectal cancer had a high risk of death and determines prognosis significantly. In our study, it is associated with inadequacy of treatment.”

Comment: In the conclusion section, authors report incorporation of screening and early detection are crucial to "increasing survival time". Screening and early detection were not included in this analysis; hence this is an inappropriate conclusion to make within the context of this current research.

Response: We agree with the reviewer comment. The conclusion now reads:

‘‘This study estimates a low overall five-year survival rate. Majority of the patient’s s were young adults and diagnosed with advanced cancer stage. Patient’s survival is largely influenced by advanced stage at presentation and delayed in the administration of adjuvant therapy. Primary tumour site, lymph node involvement, distant metastasis and treatment modalities were important prognostic factors for colorectal cancer. Rectal cancer, node-positive, and metastatic cancer at a presentation in colorectal cancer patients decreased the survival rate. Whereas, adjuvant therapy increased survival rate. Administration of adjuvant chemotherapy through improving access to chemotherapy and radiotherapy will improve outcome.” 

Comment: Several English and grammatical errors were also noted through the manuscript

Response: The paper is checked again to improve English and grammatical errors.

---

## [Decision Letter · Decision Letter 1]

17 Dec 2020

PONE-D-20-03951R1

Prognosis of Colorectal Cancer in Tikur Anbessa Specialized Hospital, the Only Oncology Center in Ethiopia.

PLOS ONE

Dear Dr. Etissa,

Thank you for submitting your manuscript to PLOS ONE. After careful consideration, we feel that it has merit but does not fully meet PLOS ONE’s publication criteria as it currently stands. Therefore, we invite you to submit a revised version of the manuscript that addresses the points raised during the review process.

We look forward to receiving your revised manuscript.

Kind regards,

Ludmila Vodickova, M.D., PhD

Academic Editor

PLOS ONE

Additional Editor Comments (if provided):

Please see attachment.

Reviewers' comments:

Reviewer's Responses to Questions

**Comments to the Author**

1. If the authors have adequately addressed your comments raised in a previous round of review and you feel that this manuscript is now acceptable for publication, you may indicate that here to bypass the “Comments to the Author” section, enter your conflict of interest statement in the “Confidential to Editor” section, and submit your "Accept" recommendation.

Reviewer #1: All comments have been addressed

Reviewer #3: All comments have been addressed

2. Is the manuscript technically sound, and do the data support the conclusions?

Reviewer #1: Yes

Reviewer #3: Yes

3. Has the statistical analysis been performed appropriately and rigorously? 

Reviewer #1: I Don't Know

Reviewer #3: Yes

4. Have the authors made all data underlying the findings in their manuscript fully available?

Reviewer #1: Yes

Reviewer #3: Yes

5. Is the manuscript presented in an intelligible fashion and written in standard English?

Reviewer #1: Yes

Reviewer #3: Yes

6. Review Comments to the Author

Reviewer #1: (No Response)

Reviewer #3: Overall authors have addressed the concerns raised by reviewers.

They did clarify the influence of lymph nodal status on decision to administer chemotherapy for stage 2 disease

7. PLOS authors have the option to publish the peer review history of their article (what does this mean?). If published, this will include your full peer review and any attached files.

Reviewer #1: No

Reviewer #3: No

---

## [Author Response · Author response to Decision Letter 1]

19 Dec 2020

Comments from the editor

Section Material and Methods:

The sentence:

„Thirteen patients not meeting the exclusion criteria were excluded from the study.“

Should sound as 

„Thirteen patients not meeting the inclusion criteria were excluded from the study.“

Response: Thank you. We have amended the text to make a sound.

Section: Sample size consideration:

Citation: Freeman 1982 is not included to the References list

Symbols as: Zα/2, Zβ, s(t), P…….. are not explained or commented.

Response: The suggested correction have been made in the specified section. Now it reads:

Sample size consideration

Sample size determination started by determining d (number of events) because of the number of observed events matters beyond the total number of patients in survival data analysis. The required total number of events (Freedman 1982) was calculated using:[20]

d =that Δ=hazard ratio is constant in time. For the two-sided log-rank test (H0: Δ = 1 vs HA: Δ ≠ 1).

Then, the required total number of patients can be calculated as A previous similar study conducted in Ghana showed that survival function (s (t)) = 0.16, thus we used. We also know that.

To obtain the total sample size required to yield a power of 90%, a hazard ratio of 1.832 and 5% two-sided level of significance, the required number of events is:

=

Zα/2 = 1.96 for 95% confidence and Zβ = probability of committing type II error to give power of 90%.

Accordingly, 

Adjusting for a loss of 20%. .

The required sample size was 172. However, all colorectal cancer patients treated in TASH from 2012 to 2016 and who fulfil the inclusion criteria were included in the study. Hence the sample used for this study was 422.

Table 1:

Family history - of (any) cancer or of only CRC?

Response: Thank you for pointing this out. Amended as Family history of CRC.

Table 2, Table 4 and Table 5

Size and location of the tumor

Should be only Size of the tumor (no info about location, relevant data about location are under „Primary tumor site“).

Response: As requested, the correction has been made throughout the manuscript.

Section: Result of multivariable Cox regression model

Unify the style of writing of capital letters in list of variables over the whole section. In this list is again idiom „size and location of tumor“– should be only „size of tumor“. 

Response: Thank you. We have made a uniform style of writing capital letters.

Section: Discussion

Line 323: unify style of writing capital letters for variables

Response: The suggested correction has been made.

---

## [Editor Report · Decision Letter 2]

20 Jan 2021

Prognosis of Colorectal Cancer in Tikur Anbessa Specialized Hospital, the Only Oncology Center in Ethiopia.

PONE-D-20-03951R2

Dear Dr. Etissa,

We’re pleased to inform you that your manuscript has been judged scientifically suitable for publication and will be formally accepted for publication once it meets all outstanding technical requirements.

Kind regards,

Chih-Pin Chuu, Ph.D.

Academic Editor

PLOS ONE
---

## [Editor Report · Acceptance letter]

22 Jan 2021

PONE-D-20-03951R2 

Prognosis of Colorectal Cancer in Tikur Anbessa Specialized Hospital, the Only Oncology Center in Ethiopia 

Dear Dr. Etissa:

I'm pleased to inform you that your manuscript has been deemed suitable for publication in PLOS ONE. Congratulations! Your manuscript is now with our production department. 

Kind regards, 

on behalf of

Prof. Chih-Pin Chuu 

Academic Editor

PLOS ONE